# Drug-Resistant *Aspergillus* spp.: A Literature Review of Its Resistance Mechanisms and Its Prevalence in Europe

**DOI:** 10.3390/pathogens12111305

**Published:** 2023-10-31

**Authors:** Maria Antonia De Francesco

**Affiliations:** Department of Molecular and Translational Medicine, Institute of Microbiology, University of Brescia, ASST Spedali Civili, 25123 Brescia, Italy; maria.defrancesco@unibs.it; Tel.: +39-030-399-5860; Fax: +39-030-399-6071

**Keywords:** drug, resistance, *Aspergillus*, azole

## Abstract

Infections due to the *Aspergillus* species constitute an important challenge for human health. Invasive aspergillosis represents a life-threatening disease, mostly in patients with immune defects. Drugs used for fungal infections comprise amphotericin B, triazoles, and echinocandins. However, in the last decade, an increased emergence of azole-resistant *Aspergillus* strains has been reported, principally belonging to *Aspergillus fumigatus* species. Therefore, both the early diagnosis of aspergillosis and its epidemiological surveillance are very important to establish the correct antifungal therapy and to ensure a successful patient outcome. In this paper, a literature review is performed to analyze the prevalence of *Aspergillus* antifungal resistance in European countries. Amphotericin B resistance is observed in 2.6% and 10.8% of *Aspergillus fumigatus* isolates in Denmark and Greece, respectively. A prevalence of 84% of amphotericin B-resistant *Aspergillus flavus* isolates is reported in France, followed by 49.4%, 35.1%, 21.7%, and 20% in Spain, Portugal, Greece, and amphotericin B resistance of *Aspergillus niger* isolates is observed in Greece and Belgium with a prevalence of 75% and 12.8%, respectively. The prevalence of triazole resistance of *Aspergillus fumigatus* isolates, the most studied mold obtained from the included studies, is 0.3% in Austria, 1% in Greece, 1.2% in Switzerland, 2.1% in France, 3.9% in Portugal, 4.9% in Italy, 5.3% in Germany, 6.1% in Denmark, 7.4% in Spain, 8.3% in Belgium, 11% in the Netherlands, and 13.2% in the United Kingdom. The mechanism of resistance is mainly driven by the TR_34_/L98H mutation. In Europe, no in vivo resistance is reported for echinocandins. Future studies are needed to implement the knowledge on the spread of drug-resistant *Aspergillus* spp. with the aim of defining optimal treatment strategies.

## 1. Introduction

*Aspergillus* spp. are filamentous fungi found ubiquitously in the environment, in places such as soil, decaying vegetative material, and dust [1]. Furthermore, the fungi might colonize oligotrophic water systems: more than 400 different species have been found to inhabit different water sources [2], underlining that as well as air, water might also be a potential source of the transmission of filamentous fungi [3].

In different geographical areas, climatic factors such as humidity, rainy season, and temperature influence the prevalence of *Aspergillus* spp. [4].

The inhalation of *Aspergillus* conidia gives rise to different respiratory infections, affecting immunocompromised patients more severely. 

Despite great advances in the diagnosis and treatment of aspergillosis, mortality remains high, particularly in subjects with important immune defects and invasive diseases.

Infections are generally due to *Aspergillus fumigatus*, even if other species are increasingly detected as etiological agents [5]. 

Recently, *Aspergillus fumigatus* was added to the list of the 19 fungal pathogens to be prioritized by the World Health Organization (WHO) and inserted into the critical group together with *Cryptococcus neoformans*, *Candida auris*, and *Candida albicans* [6].

Antifungal resistance is an emerging and important challenge in different parts of the world, with up to 20% of *Aspergillus* isolates displaying de novo resistance to commonly used antifungal drugs [7].

The aims of this review are to update the different antifungal-resistance mechanisms found in *Aspergillus* spp. and describe the prevalence of drug-resistant *Aspergillus* spp. in European countries to provide information for the adequate treatment of aspergillosis in this geographic area.

## 2. Summary of the Clinically Relevant Species

The genus *Aspergillus* is included in the family *Aspergillaceae*, order *Eurotiales* [8], and comprises six subgenera: *Aspergillus*, *Circumdati*, *Cremei*, *Fumigati*, *Nidulantes*, and *Polypaecilum* [9,10,11]. Every subgenus is divided into sections. In particular, two sections are included in the subgenus *Aspergillus*; 10 sections are included in the subgenus *Circumdati*; only one section is included in the subgenus *Cremei*; in the subgenus *Fumigati*, there are 4 sections; in the subgenus *Nidulantes*, there are 9 sections and, in the subgenus *Polypaecilum*, only one section is included (Figure 1) [9,10,11].

Clinically relevant *Aspergillus* spp. are mostly included in the sections *Fumigati*, *Flavi*, *Nidulantes*, *Nigri*, *Terrei*, and *Usti*.

In the *Fumigati* section, there are 63 species [12]. *Aspergillus fumigatus* [12,13] is the species of this section mostly found in different countries and associated with human infection. Other clinically relevant species found are *A. felis*, *A. fischeri*, *A. fumigatiaffinis*, *A. fumisynnematus*, *A. hiratsukae*, *A. laciniosus*, *A. lentulus*, *A. novofumigatus*, *A. pseudoviridinutans*, *A. spinosus*, *A. thermomutatus*, *A. udagawae*, and *A. viridinutans* [12,14,15]. 

In the *Flavi* section, there are 35 species, according to the new taxonomy revision, where the most prevalent species in human infections was represented by *Aspergillus flavus*, followed by nine other species [11,12,16,17,18,19,20].

In the *Terrei* section, which comprises 17 species [11], *A. terreus* is the most frequent species of the section associated with invasive aspergillosis in immunocompromised patients and is responsible for 5.2% of all fungal infections [21,22,23,24].

The section *Nigri* includes 30 species, of which only 8 were found to be responsible for invasive aspergillosis: *A. brasiliensis*, *A. carbonarius*, *A. japonicus*, *A. luchuensis*, *A. niger*, *A. tubingensis*, *A. uvarum*, and *A. welwitschiae* [10,12,25]. 

In the *Nidulantes* section, there are 74 species [12,26]. Of them, only 11 species have been associated with human infections and the most prevalent species was *A. nidulans*, identified as an etiological agent of invasive aspergillosis and chronic granulomatous disease [27,28,29,30,31,32,33].

In the *Usti* section, there are 25 species, of which *A. calidoustus* was the species most often associated with invasive infections [12].

## 3. Clinical Picture

*Aspergillus* spp. are etiological agents responsible for different clinical manifestations in humans [34,35]. The severity of infections is dependent on the immune status of the patient and can be classified into noninvasive infections and invasive infections according to host immunocompetency.

### 3.1. Noninvasive Infections in the Immunocompetent Host

#### 3.1.1. Chronic Pulmonary Aspergillosis

Chronic pulmonary aspergillosis (CPA) includes the chronic forms of aspergillosis, as well as aspergilloma, which generally affects patients with a preexisting pulmonary pathology such as tuberculosis or other cavitary chronic lung diseases [36]. The most common picture of CPA is chronic cavitary pulmonary aspergillosis (CCPA). Without effective therapy, CCPA may lead to chronic fibrosing pulmonary aspergillosis (CFPA). Aspergilloma, characterized by mycelial mass, inflammatory cells, fibrin and mucus, and *Aspergillus* nodules, is a less severe clinical picture of CPA [37].

Clinical symptoms of patients with CPA are chronic productive cough, weight loss, and hemoptysis with the presence of nodules and fungal balls at radiological observation [37].

CPA is estimated to affect 3 million people annually, but because it is often not recognized, the real incidence may be greater [37,38,39].

#### 3.1.2. Allergic Bronchopulmonary Aspergillosis

Allergic bronchopulmonary (ABPA) is caused by hypersensitivity to *Aspergillus fumigatus* and affects almost exclusively atopic individuals and patients with cystic fibrosis, affecting nearly 5 million people annually [36,38]. In particular, the rate of ABPA in patients with cystic fibrosis is estimated to be 7.8% and is different among patients with various gene mutations [40]. 

Clinical symptoms are chronic cough, wheezing, low-grade fever, chest pain, blood eosinophilia, and central bronchiectasis on chest imaging.

### 3.2. Invasive Infections in the Immunocompromised Host

Invasive aspergillosis (IA) is the most severe clinical picture of aspergillosis, occurring in severely immunocompromised hosts [36]. The respiratory tract is the most common primary site of invasive aspergillosis, but because *Aspergillus* hyphae may invade pulmonary arterioles, a hematogenous spread with thrombosis, hemorrhagic infarction, and invasion of distant organs such as kidneys, liver, spleen, sinuses, and the central nervous system might occur in about 25% of patients. 

Other clinical manifestations of IA, although less common, are osteomyelitis, arthritis, or subacute thyroiditis [9]. Endophthalmitis, secondary to intraocular surgery or hematogenous dissemination, is related to poor ocular prognosis [41]. The incidence of IA, which is estimated to affect 250,000 people globally, is still growing [38]. This rise is potentially due to the increased use of immunosuppressive treatments for cancer, hematological tumors, stem cell transplants, and solid organ transplants responsible for neutropenia in affected patients. Neutrophil recruitment and the production of reactive oxygen species play an important role in the inhibition of the germination of *Aspergillus* conidia. Therefore, severe and prolonged neutropenia is considered a major risk for IA. However, a shift in incidences of IA from neutropenic to non-neutropenic patients was recently observed, in particular in patients with viral infections such as influenza and SARS-CoV-2 [36,42] and in patients with CD4+ T-cell dysfunction [43].

It has also been found that invasive pulmonary aspergillosis shows different pathophysiological mechanisms based on the type of immunosuppression. Angio-invasive manifestation is principally detected in patients with neutropenia, while non-angio-invasive form was found in patients with corticosteroid-induced immunosuppression [44,45].

When IA is refractory to the therapy, the mortality rates are higher, ranging from 50% to 100% [46].

## 4. Diagnosis and Therapy

The improvement of disease outcomes in patients at higher risk of developing invasive aspergillosis is strictly associated with rapid diagnosis. To date, because there is not a specific assay, a combination of clinical, radiological, and microbiological features is recommended for an accurate diagnosis of invasive aspergillosis [47,48,49].

Routine microbiological tests include direct microscopic examination and culture with limited sensitivity that cannot alone discriminate between colonization and infection; furthermore, prior therapy is associated with false-negative culture results [50,51]. 

Other microbiological approaches are serological tests targeting the cell-wall component galactomannan (GM) in serum and bronchoalveolar fluid (BALF), the detection of the fungal cell-wall component 1, 3-β-D glucan in serum (although nonspecific for *Aspergillus* diseases only), the detection of *Aspergillus*-specific siderophores in BALF and urine, the detection of GM by a lateral flow device assay (LFD), and molecular tests such as PCR [52,53,54,55,56,57,58,59].

To overcome the limited sensitivity of any individual tests, combining different diagnostic assays is the currently recommended procedure because it has been shown that there is a significant increase in sensitivity with a slight reduction of specificity [60].

The antifungal drugs currently used for the therapy of aspergillosis comprise three classes of compounds, two of which target ergosterol (triazoles and amphotericin B), and a third class (echinocandins) that arrests the synthesis of beta-1,3 glucan, an important constituent of the cell wall [61,62,63,64,65]. However, allergic forms of infection may need glucocorticoids or anti-IgE therapy, as well as antifungal drugs, while aspergilloma can be treated by surgery.

Voriconazole, isavuconazole, and posaconazole, belonging to the triazoles class, are the first-line agents for invasive infections, while voriconazole or itraconazole are the first-line agents for chronic diseases [66,67,68].

For the management of refractory aspergillosis, a combination therapy with drugs that have different mechanisms of action, such as voriconazole or amphotericin B and an echinocandin, is the suggested approach [69]. Resistance to first-line agents among *Aspergillus* spp. has given rise to the elevated utilization of second-line agents represented by echinocandins in monotherapy [70]; these drugs were initially approved by the FDA for the management of invasive aspergillosis refractory to the standard therapy [71].

Antifungal susceptibility testing to assess antifungal drug activity is performed by broth microdilution tests or gradient strip tests according to the guidelines of the European Committee on Antimicrobial Susceptibility Testing (EUCAST) or the Clinical and Laboratory Standards Institute (CLSI) [72]. For *Aspergillus* spp., in contrast to *A. fumigatus*, because of the absence of clinical breakpoints, epidemiological cut-off values (ECOFFs) are available from the EUCAST for the most used drugs (triazoles) [73].

## 5. Antifungal-Resistance Mechanisms

The emergence of resistance to antifungals affects their clinical effectiveness, leading to a great public health problem worldwide. Here, we summarize all mechanisms underlying antifungal resistance in *Aspergillus* spp. 

### 5.1. Amphotericin B Resistance

Amphotericin B acts by interacting with sterols, in particular with ergosterol, the principal component of the fungal cell membrane. The integration of amphotericin B into the fungal membrane leads to the formation of channels (Figure 2). This formation impairs barrier membrane function, increasing the permeability responsible for the leakage of potassium, protons, cations, and cytoplasmic materials that induce cell death [74]. Furthermore, amphotericin B can produce reactive oxygen species (ROS) that give rise to cellular damage [75]. Among *Aspergillus* spp., *Aspergillus terreus* is the strain harboring intrinsic resistance to amphotericin B [76].

The mode of action of amphotericin B in *Aspergillus terreus* has not been well elucidated, and no genomic features have been identified to date that might be linked to amphotericin B resistance. However, amphotericin B resistance in section *Terrei* seems to be associated with the modulation of molecular chaperones, targeting ROS by mitochondria and influencing cellular redox homeostasis, with an increase in the level of catalase and superoxide dismutase with respect to other *Aspergillus* species [76,77,78] (Figure 2).

Following the increasing rate of azole resistance [79], the use of amphotericin B has enhanced, and this might be the reason why, recently, an increase in MIC values for amphotericin B in different *Aspergillus* species has been reported [80,81,82,83], even if resistance to this drug remains extremely rare [84].

In a recent study [85], among 26,909 *Aspergillus* isolates analyzed, resistance to amphotericin B was detected in 36.8% of *Aspergillus terreus*, 14.9% of *Aspergillus flavus*, 5.2% of *Aspergillus niger*, and 2.01% of *Aspergillus fumigatus* isolates. Furthermore, some *Aspergillus lentulus* and *Aspergillus ustus* isolates have been reported to show amphotericin B resistance [86,87]. Additionally, an increasing trend in amphotericin B resistance was observed in *Aspergillus fumigatus* between 2016 and 2020, together with a decreasing trend in amphotericin B resistance in *Aspergillus terreus* and *Aspergillus flavus* [85].

### 5.2. Azole Resistance

Azole antifungal drugs act by interfering with the synthesis of ergosterol, mediated by the fungal sterol 14 alpha-demethylases (Cyp51A and Cyp51B) (Figure 3). 

The principal mechanisms of azole resistance involve (a) mutations in the sterol-demethylase gene *cyp51A* reducing the affinity between the azole drug and its target; (b) the overexpression of the sterol-demethylase gene *cyp51A* leading to an increase of the azole concentration able to inhibit fungal growth; (c) the overexpression of efflux pump systems decreasing the intracellular drug concentration [88,89]. Alternate mechanisms associated with azole resistance are the modification of HapE [90], the involvement of the mitochondrial complex 1 and the cytochrome b_5_-CybE redox systems [91,92], the transcription factors SrbA and AtrR [93,94] and biofilm formation (Figure 3) [95].

The acquisition of azole resistance occurs in two ways: in vivo, by selection of resistant isolates during long therapy with azoles, and in vitro, by the selection of resistant isolates as a consequence of an extensive use of azole fungicide in agriculture [96,97,98]. 

#### 5.2.1. Mutations in the Sterol-Demethylase Gene Cyp51

*Cyp51* genes are the major targets studied for azole resistance in fungal pathogens—resistance that might also be acquired by horizontal gene transfer (HGT) [99]. Different species of *Aspergillus* exhibit several numbers of *cyp51* paralogs in their genome. This genetic redundancy gives the fungus an advantage to survive when exposed to fungicides, increasing its azole resistance [100]. In *Aspergillus fumigatus*, *Aspergillus terreus*, and *Aspergillus niger*, there are only two paralogs (*cyp51A* and *cyp51B*), while in *Aspergillus flavus*, three paralogs exist (*cyp51A*, *cyp51B* and *cyp51C*) [101]. However, only specific mutations in Cyp51A and Cyp51C proteins have been shown to have an impact on azole resistance, while the Cyp51B protein might play other roles that need to be studied in detail [102].

In *Aspergillus fumigatus*, amino-acid mutations in the Cyp51A protein related to azole resistance were G54, Y121, G138, P216, F219, M220, A284, Y431, G432, G434, and G448 [103,104,105,106,107,108,109]. No mutations were associated with azole resistance in the CypB protein [88,110]. 

Also, in *Aspergillus lentulus*, it was demonstrated by targeted *cyp51A* gene knockout that intrinsic azole resistance was related to this gene [111].

In *Aspergillus flavus* with reduced voriconazole susceptibility, different mutations were reported in Cyp51A, Cyp51B, and Cyp51C proteins, even if their role in azole resistance needs to be clarified by further studies. Amino-acid changes in Cyp51A protein were identified at positions R450S, K197N/D282E/M288L, and Y132N/T469S [112]; in Cyp51B protein, detected mutations were H399P, D411N, T454P, T486P 105, and Q354K [113]; in Cyp51C protein, the Y319H mutation was identified from an azole-resistant clinical isolate [114] and the mutations S196F, A324P, N423D, and V465M [115,116].

In section *Nigri*, in Cyp51A protein, many mutations were identified, but their role in inducing azole resistance is still uncertain [117].

In particular, in *Aspergillus tubingensis*, a recent study found the amino-acid change H467Q in combination with the mutations K64E or V377I exclusively in non-wildtype isolates [118], while in *Aspergillus niger*, no mutation is associated with azole resistance [118].

However, it has been found that single-gene deletions of *cyp51A* and *cyp51B* genes in *Aspergillus tubingensis* and *Aspergillus niger* decrease the voriconazole MIC values below the ECV established by CLSI [119].

In *Aspergillus braziliensis* with reduced azole susceptibility, there were no mutations present in Cyp51A protein [118].

Regarding the *Terrei* section, few studies are available in the literature about their azole resistance; only a mutation of methionine in position 217 in *Aspergillus terreus* has been reported [120,121].

In the *Usti* section, the intrinsically azole-resistant species *Aspergillus calidoustus* exhibited a mutation M220V in Cyp51A protein, a position already associated with azole resistance in *Aspergillus fumigatus* [88].

#### 5.2.2. Overexpression of the Sterol-Demethylase Cyp51

The overexpression of Cyp51 is considered another mechanism of azole resistance in *Aspergillus* spp., in particular in *Aspergillus fumigatus*. 

Regarding *Aspergillus flavus*, the overexpression of Cyp51A and Cyp51B was not related to azole resistance because the levels of gene expression were the same both in wild and non-wildtype strains [122,123]. Also, in the *Nigri* section, this mechanism seems not to be related to azole resistance, even if Cyp51A is upregulated after azole exposure [118,124]. 

Changes in the promoter region of *cyp51* have been described as a mechanism to contrast azole toxicity, mostly in *Aspergillus fumigatus*, such as the insertion of tandem repeats (TR) of 34bp, 46bp, and 53 bp leading to upregulation of *cyp51* [125]. These insertions were often associated with amino-acid substitutions in Cyp51A and have been observed in strains that exhibit reduced susceptibility to azoles [126].

In particular, the TR_34_/L98H mutation was found principally in resistant environmental isolates, linked to total resistance to itraconazole and reduced susceptibility to voriconazole and posaconazole [89,127,128].

Furthermore, the TR_46_/Y121F/T289A mutation was found in isolates with high levels of resistance to voriconazole and other azoles [129]. For the tandem repeat of 53 bp, no amino-acid substitution has been found to date [128,130].

Other transcription factors were found able to regulate cyp51 expression in *Aspergillus fumigatus*, such as SrbA, HapE, and AtrR.

SrbA is a transcriptional regulator that is involved in different processes, as well as sterol biosynthesis [131,132,133], and its deletion leads to greater azole susceptibility [93].

The mutation P88L in heme activator protein E (HapE), a subunit of the CCAAT-binding transcription factor complex (CBC), determines an upregulation of the *cyp51A* gene and confers azole resistance [134]. A decrease in CBC activity, a negative regulator of the ergosterol pathway, increases the expression of the enzymes involved in its biosyntheses, such as HMG-CoA-synthase, HMG-CoA-reductase, and sterol C14-demethylase contributing to azole resistance [90].

Furthermore, a Zn2Cys6 cluster-containing transcription factor, called ABC transporter regulator or AtrR, was found to be important for azole tolerance both in *Aspergillus fumigatus* and *Aspergillus flavus* [135,136]. Additionally, cytochrome b5CybE has been found to be able to regulate the expression level of *cyp51A* [92].

Furthermore, biofilm production was also hypothesized to play a role in the azole resistance of *Aspergillus fumigatus*. The cell density reached in a mature biofilm, together with the production of polysaccharide extracellular matrix, protects the mold by the action of the immune system and the antifungal drugs [96].

#### 5.2.3. Overexpression of Efflux Pump Systems

Efflux pumps are transmembrane proteins that expel drugs from the cell, reducing their intracellular concentration. Therefore, the overexpression of these proteins might be related to azole resistance. The principal types of efflux pumps are the ATP-binding cassette (ABC) transporter and the major facilitator superfamily (MFS) transporter, which differ in structure and activity [137]. ABC transporters use ATP as an energy source, while MFS transporters use a proton gradient to pump out the drugs [138].

The more studied ABC transporters involved in azole resistance are *cdr1B*, *mdr1*, *mdr2*, *mdr3*, *mdr4*, *abcD*, *abcE*, *atrI*, *atrB*, *atrC*, and *atrF.* In *Aspergillus fumigatus* and *Aspergillus flavus*, only the ABC transporter Cdr1B has been found to be related to azole resistance [139,140].

MFS transporters were less studied, and among them, only *mdrA* has been associated with an increase of itraconazole and voriconazole susceptibility in *Aspergillus fumigatus* [141].

### 5.3. Echinocandin Resistance

Echinocandins (caspofungin, anidulafungin, and micafungin) act by inhibiting the glucan synthase, an enzyme codified by the *FKS1* and *FSK2* genes, important for the synthesis of the beta1, 3 glucan (Figure 4). However, because echinocandins have only a fungistatic activity against *Aspergillus* spp., they are used only in combination with a polyene or an azole to obtain an important synergistic effect. To date, the echinocandin resistance is rarely found in *Aspergillus* spp. [142].

Some studies have reported mutations in *FSK* genes [143] and changes in the lipid profile around the enzyme [144] as possible mechanisms of echinocandin resistance (Figure 4). In *Aspergillus fumigatus*, two mutations (S678P and E671Q) in the *FSK1* gene were associated with echinocandin resistance [145,146], while in *Aspergillus flavus* it was shown that P-type ATPase and ubiquinone biosynthesis methyltransferase COQ5 might be involved in caspofungin resistance [147].

## 6. Prevalence of *Aspergillus* spp. Drug Resistance in Europe

### 6.1. Amphotericin B Resistance

Amphotericin B resistance is rarely represented in *Aspergillus fumigatus*, while poor susceptibility to this drug was reported for *Aspergillus terreus*, *Aspergillus flavus*, and *Aspergillus niger* [85].

In distinct geographical areas, prevalence of these molds was different: amphotericin B-resistant *Aspergillus niger* isolates were found at higher frequencies in Asia and America (20.9% and 2.7%, respectively) than in Europe (0.62%) [85], while for amphotericin B-resistant *Aspergillus terreus* isolates, a lower prevalence was found in America (25.1%) than Asia and Europe (40.4% and 40.1%, respectively) [85].

In European countries, the prevalence of amphotericin B-resistant *Aspergillus* species was quite different (Figure 5). In France, a prevalence of 84% (31/37) isolates of *Aspergillus flavus* was reported [81], while in Spain, 38/77 (49.4%) isolates of *Aspergillus flavus* were amphotericin B-resistant 80. In Italy, a different prevalence of amphotericin B was reported between distinct *Aspergillus* species, with 5/13 (38.5%) isolates of *Aspergillus terreus*, 2/10 (20%) isolates of *Aspergillus flavus* and 6/38 (15.7%) isolates of *Aspergillus oryzae* [148].

In Greece, a total prevalence of amphotericin B resistance of 17.6% (18/102) was found, which was distributed as follows: 21.7% in *Aspergillus flavus* isolates (5/23), 10.8% in *Aspergillus fumigatus* isolates (4/37), 75% in *Aspergillus niger* isolates (3/4), 33.3% in *Aspergillus terreus* isolates (4/12), and 22.2% in *Aspergillus nidulans* isolates (2/9) [149]. In Portugal, a prevalence of amphotericin B resistance of 35.7% in *Aspergillus flavus* (5/14) was reported, while all the *Aspergillus lentulus*, *Aspergillus terreus*, and *Aspergillus felis* isolates were resistant to this drug [150].

In Belgium, amphotericin B resistance had a prevalence of 12.8% in *Aspergillus* section *Nigri* isolates [151], and in Denmark, 3/112 isolates of *Aspergillus fumigatus* (2.6%) exhibited amphotericin B resistance [152].

The accurate establishment of amphotericin B resistance is difficult because different MIC assays might be used in different laboratories. Moreover, there is a lack of breakpoints for *Aspergillus* species different from *Aspergillus fumigatus*, and changes in amphotericin B susceptibility breakpoints have been reported.

### 6.2. Azole Resistance

Many studies have principally focused on azole-resistant *Aspergillus fumigatus* isolates because they represent the predominant pathogen of aspergillosis. The overall azole resistance rate of *Aspergillus fumigatus* was reported as ranging from 0.6 to 27.8%, depending on the isolation country, the type of disease, and the emergence of the environmental resistance mechanism [153].

Most of the environmental azole-resistant isolates were found in Europe (56.7%) than in other countries due to the higher azole fungicide application per hectare of agricultural land [39].

The Netherlands was the European country using the greatest amount of azole fungicide, followed by Germany and France. In fact, the first environmental pan-azole-resistant *Aspergillus fumigatus* isolate was detected in the Netherlands [154].

The most recent studies found in the literature have reported the prevalence of azole resistance in *Aspergillus* spp. from Austria, the United Kingdom, Belgium, France, Denmark, Portugal, the Netherlands, Greece, Spain, Switzerland, Italy, and Germany (Figure 6).

A survey conducted in Tyrol, Austria, showed a low prevalence (1/388 isolates, 0.29%) of azole resistance of *Aspergillus fumigatus*, while *Aspergillus terreus* had a percentage of resistance to posaconazole ranging from 0.3% during the period 2007–2009 to 0.6% during the period 2010–2012 fading away from 2013 to 2017 [155].

In the United Kingdom, analysis of urban and rural environments showed a prevalence of azole-resistant *Aspergillus fumigatus* of 6.7%, reaching a prevalence of 13.8% in urban environments. The TR_34_/L98H mutation was the most detected one. The TR_46_/Y121F/T289A resistance allele was also identified for the first time in this country [156].

For clinical isolates, a study performed in a cardiothoracic center in London detected a higher prevalence of azole-resistant *Aspergillus fumigatus* (13.2%) principally associated with the environmentally driven TR_34_/L98H mutation [157].

In Belgium, the prevalence of azole resistance of *Aspergillus fumigatus* isolates analyzed in a hospital in Leuven was found to be stable during the study period (2016–2020) ranging from 8.3% in 2016 to 7.4% in 2020, with an overall five-year azole resistance prevalence of 7.1%. The TR_34_/L98H mutation was the most predominant (83%), followed by the TR_46_/Y121F/ T289A (13.8%) [158].

A recent study performed in Lyon, France [159] detected a prevalence of azole resistance of 2.1% (4/195) of *Aspergillus fumigatus* isolates, which increased to 4.3% in patients with cystic fibrosis, a percentage similar to that found in Paris [160] and similar to that found in Rennes (5% of isolates resistant to voriconazole) [161], but lower than that observed in Nantes (6.8%) [162] for the same category of patients. One azole-resistant isolate exhibited the F46Y, M172V, and 427K mutations in the *cyp51A* gene. No mutations were identified in the *cyp51A* promoter, but significant induction of *cyp51A*, *cyp51B*, *atrF*, and *cdr1B* gene expression was observed in the resistant isolates.

In Denmark, an azole resistance prevalence of 6.1% was observed during the nationwide surveillance study period (2018–2020), with a TR_34_/L98H mutation identified in 3.6% of patients, while the TR_46_/Y121F/T289A mutation was not detected [163]. Furthermore, in the Danish population affected by cystic fibrosis, an azole prevalence of 7.3%, including 3.7% TR_34_/L98H, was observed. 

Furthermore, in Denmark, a fatal clinical case of infection with an isolate of *Aspergillus fumigatus* that acquired an unusual 120-bp tandem repeat resistance mechanism during long-term azole treatment was recently described [164].

In Portugal, a survey performed in three hospitals showed a prevalence of isolates belonging to *Aspergillus fumigatus* complex (86.7%) and 7.5% of isolates belonging to cryptic species. The latter presented percentages of 47.1%, 82.4%, and 100% of voriconazole, posaconazole, and isavuconazole resistance, respectively [137]. Eight *Aspergillus fumigatus* sensu stricto isolates showed azole resistance (8/227, 3.5%). Three isolates of them with pan-azole-resistant profiles exhibited the TR_34_/L98H and TR_46_/Y121F/T289A mutations; the other three azole-resistant ones showed F46Y/M172V/N248T/D255E/E427K mutations, while the other two isolates carried no mutations.

During the 6-year study period (2013–2018), the prevalence of azole resistance of *Aspergillus fumigatus* isolates in the Netherlands was 11% (508/4496), rising from 7.6% in 2013 to 14.7% in 2018 [165]. The predominant detected mutations were TR_34_/L98H (69%) and TR_46_/Y121/T289A (17%).

A soil-sampling survey was conducted in Greece [166] to analyze the prevalence of environmental azole-resistant *Aspergillus* spp. showing that *Aspergillus niger* was the most frequent species complex (57%), followed by *Aspergillus terreus* (17%), *Aspergillus fumigatus* (14%), and *Aspergillus flavus* (5%). All non-fumigatus *Aspergillus* species exhibited full susceptibility to azoles, while only one isolate (1/101, 1%) of *Aspergillus fumigatus* was pan-azole-resistant, carrying the TR_46_/Y121/T289A mutation in the *cyp51A* gene.

In Spain, the prevalence of azole resistance observed in *Aspergillus fumigatus* sensu lato was 7.4% (63/847). Azole resistance was higher in cryptic species (18/19, 95%) than in *Aspergillus fumigatus* sensu stricto (45/828, 5.5%) and principally associated with the TR_34_/L98H mutation (24/63, 38%) [167].

A screening of 160 clinical samples performed to detect azole-resistant *Aspergillus fumigatus* isolates in the Hospital of Geneva, Switzerland, reported only two pan-azole-resistant isolates (1.2%) with the TR_34_/L98H mutation [168].

A multicenter study performed in Italy reported an overall azole resistance prevalence of 4.9% and 6.9%, considering only the *Aspergillus fumigatus* sensu stricto isolate associated with the TR_34_/L98H mutation [169]. Furthermore, the F46Y/M172V/N248T/D255E/E427K amino-acid changes were observed in one azole-resistant isolate.

In Germany, a multicenter study detected a prevalence of azole-resistant *Aspergillus fumigatus* isolates of 5.3% (51/961) from patients with cystic fibrosis, and the most frequent mutation was TR_34_/L98H [170]. 

### 6.3. Echinocandin Resistance

In the literature, there were no reports about the echinocandin resistance of *Aspergillus* spp. isolates in Europe. Only a clinical isolate of *Aspergillus fumigatus*, recovered from a patient on micafungin therapy for chronic pulmonary aspergillosis, with an echinocandin resistance-associated point mutation in the well-conserved hot-spot 1 region of *fks1* conferring an F675S amino-acid substitution, was reported [143].

## 7. Conclusions

Our review underlines that, to date, in European countries, amphotericin B resistance is rarely detected in *Aspergillus fumigatus* isolates. By contrast, the emergence of azole-resistant *Aspergillus fumigatus* isolates represents an important concern in Europe, with countries from Austria, which reported the lowest prevalence (0.3%), to the United Kingdom and the Netherlands, which presented the highest prevalence (13.2% and 11%, respectively), increasing in patients with cystic fibrosis. Furthermore, the principal mechanism of azole resistance relies on the TR34/L98H mutation, which comprises many of the resistant isolates. This allele was the first to be linked to pan-azole resistance, and it has been related to the overuse of environmental azoles, underlining the need to limit the use of azoles in agriculture.

On the other side, the frequency of invasive fungal disease caused by other non-fumigatus *Aspergillus* species is increasing. Furthermore, the treatment of these invasive fungal diseases is complicated by their often intrinsic resistance to amphotericin B and the exhibition of various high MICs against triazoles.

Therefore, susceptibility testing of clinically important isolates has been strongly recommended in the European Society of Clinical Microbiology and Infectious Diseases (ESCMID) aspergillosis guidelines, suggesting a combination therapy until a susceptibility pattern is known that can guide towards a targeted systemic antifungal treatment.

These data highlight the requirement of continuous surveillance to monitor the frequency of *Aspergillus* species involved in fungal diseases in different geographical areas and the occurrence of antifungal drug resistance to start with the appropriate therapy as soon as possible for a successful patient outcome.

## Figures and Tables

**Figure 1 pathogens-12-01305-f001:**
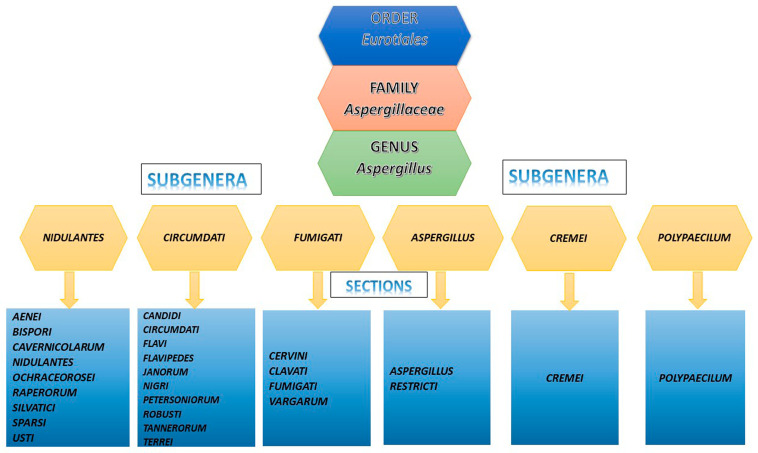
Classification of *Aspergillus* spp.

**Figure 2 pathogens-12-01305-f002:**
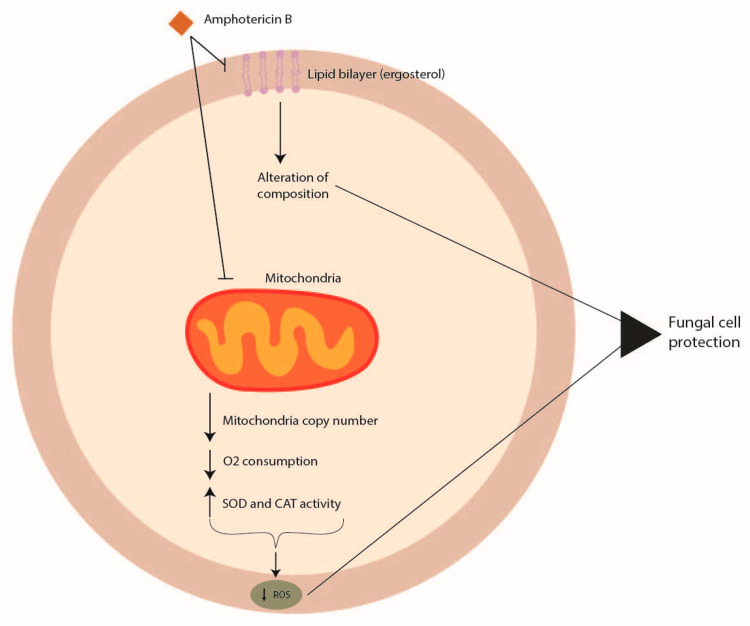
Mechanisms of *Aspergillus* spp. antifungal drug resistance: amphotericin B resistance.

**Figure 3 pathogens-12-01305-f003:**
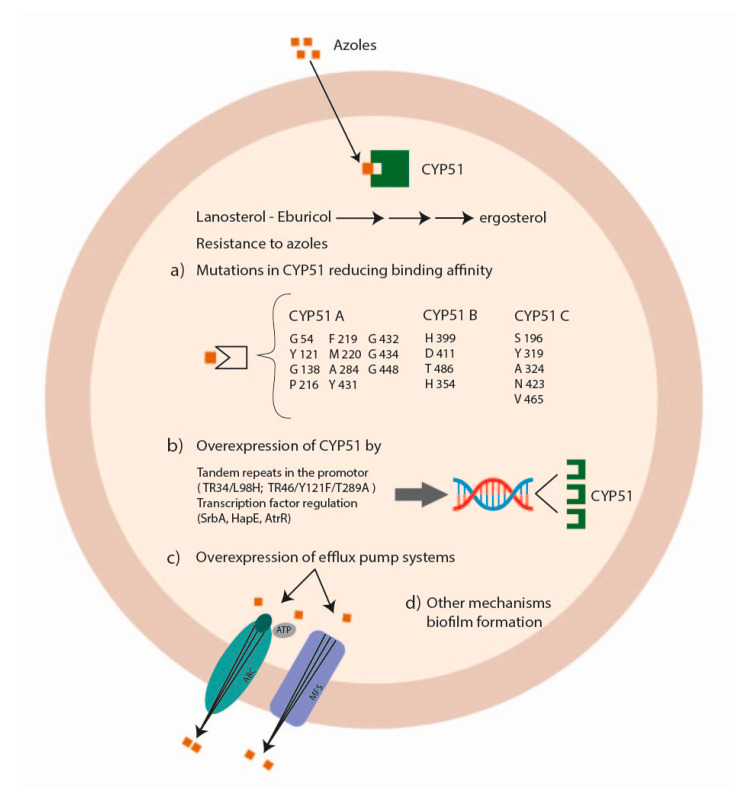
Mechanisms of *Aspergillus* spp. antifungal drug resistance: azole resistance.

**Figure 4 pathogens-12-01305-f004:**
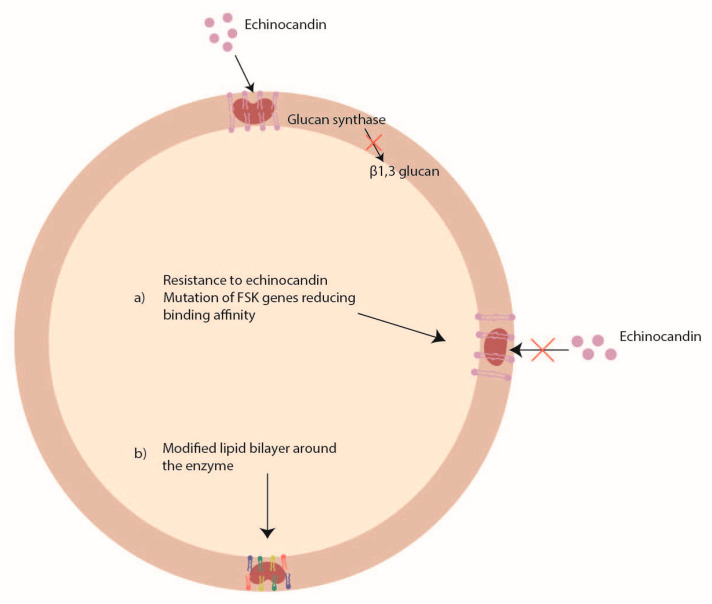
Mechanisms of *Aspergillus* spp. antifungal drug resistance: echinocandin resistance.

**Figure 5 pathogens-12-01305-f005:**
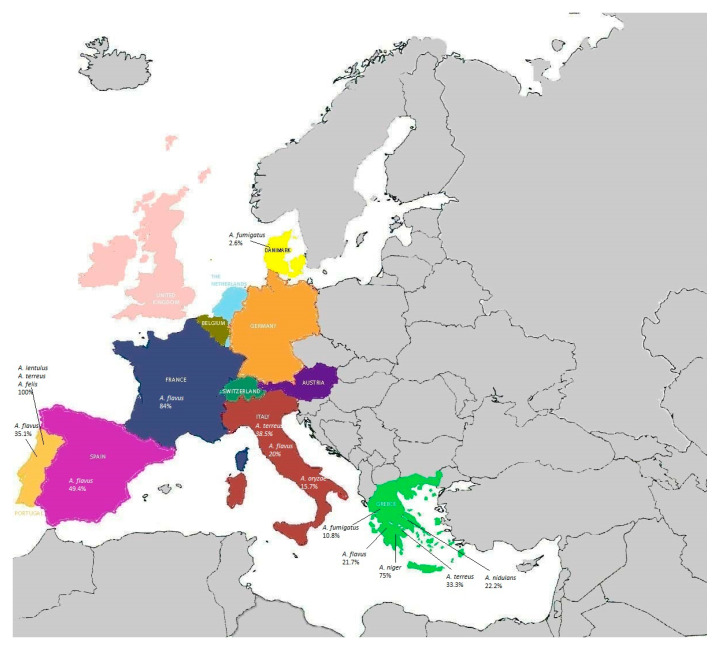
Prevalence of amphotericin B-resistant *Aspergillus* spp. in European countries.

**Figure 6 pathogens-12-01305-f006:**
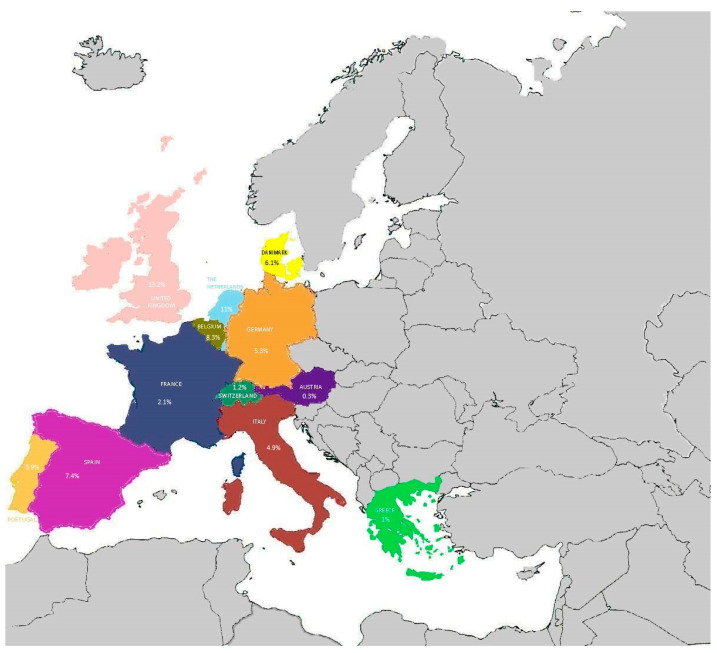
Prevalence of azole-resistant *Aspergillus fumigatus* strains in European countries.

## Data Availability

Not applicable.

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
