# Peer review of "Drug-Resistant Aspergillus spp.: A Literature Review of Its Resistance Mechanisms and Its Prevalence in Europe"

_pathogens, 2023, doi:10.3390/pathogens12111305_

Round 1
Reviewer 1 Report
Comments and Suggestions for Authors
The author has provided a comprehensive literature review on the subject of drug resistance in Aspergillus species, with a primary focus on A. fumigatus and its prevalence in Europe. The review is commendably well-written, making it both informative and enjoyable to read. However, I would like to kindly suggest a few minor points for consideration:
-
1. The review appears to have been completed in early 2022. It would be beneficial to incorporate some more recent and influential studies, such as Rhodes et al. 2022 (Nat. Microbiol), Morogovsky et al. 2022 (Microbiol Spectr), Guegan et al. 2021 (Front Cell Infect Microbiol), and the latest findings from Current Fungal Infection Reports in 2023, among others.
-
2. It would be highly appreciated if the author could include a citation of the World Health Organization's recent priority list of fungal pathogens affecting humans.
-
3. I would like to kindly propose a reorganization of the items in Figure 1. This rearrangement would serve to optimize space utilization, improve the clarity of subgenera and sections, and enhance overall visual representation.
Author Response
Dear Editor and Reviewers
Thank you for giving me the opportunity to revise this paper, for the interest in this work and the helpful suggestions
Reviewer 1
The author has provided a comprehensive literature review on the subject of drug resistance in Aspergillus species, with a primary focus on A. fumigatus and its prevalence in Europe. The review is commendably well-written, making it both informative and enjoyable to read. However, I would like to kindly suggest a few minor points for consideration:
The review appears to have been completed in early 2022. It would be beneficial to incorporate some more recent and influential studies, such as Rhodes et al. 2022 (Nat. Microbiol), Morogovsky et al. 2022 (Microbiol Spectr), Guegan et al. 2021 (Front Cell Infect Microbiol), and the latest findings from Current Fungal Infection Reports in 2023, among others.
All the suggested references were added except the last one for the not availability to access the paper
It would be highly appreciated if the author could include a citation of the World Health Organization's recent priority list of fungal pathogens affecting humans.
The citation and the respective reference have been added
I would like to kindly propose a reorganization of the items in Figure 1. This rearrangement would serve to optimize space utilization, improve the clarity of subgenera and sections, and enhance overall visual representation.
The Figure 1 has been reorganized
Reviewer 2 Report
Comments and Suggestions for Authors
The first part of the article has some educational value, providing general information on clinically relevant species, including complex taxonomic relationships, pathogenicity, and mechanisms of resistance to commonly used antifungal drugs. Of educational value are the numerous and very clear figures. The second part of the paper is an overview of the epidemiology of Aspergillus drug resistance in Europe. It is unfortunate that the authors did not include all countries belonging to the EU community in the analysis. The authors' definition of Europe was left without the slightest comment.
Author Response
Dear Editor and Reviewers
Thank you for giving me the opportunity to revise this paper, for the interest in this work and the helpful suggestions
Reviewer 2
The first part of the article has some educational value, providing general information on clinically relevant species, including complex taxonomic relationships, pathogenicity, and mechanisms of resistance to commonly used antifungal drugs. Of educational value are the numerous and very clear figures. The second part of the paper is an overview of the epidemiology of Aspergillus drug resistance in Europe. It is unfortunate that the authors did not include all countries belonging to the EU community in the analysis. The authors' definition of Europe was left without the slightest comment.
We agree with the Reviewer’s comment, but unfortunately there was not literature, to the best of my knowledge, for all the European countries.
Reviewer 3 Report
Comments and Suggestions for Authors
Excellent review. Well done! I do miss one point which could be briefly addressed if the author would like to do so: the possible role of water aerosols in the aspiration of conidia such as the one reported by Warris et al. Molecular epidemiology of Aspergillus fumigatus isolates recovered from water, air, and patients shows two clusters of genetically distinct strains. J. Clin. Microbiol. 2003, 41, 4101–4106 and addressed by Babič, M.N. et al. Fungal Contaminants in Drinking Water Regulation? A Tale of Ecology, Exposure, Purification and Clinical Relevance. Int. J. Environ. Res. Public Health 2017, 14, 636. https://doi.org/10.3390/ijerph14060636
Author Response
Dear Editor and Reviewers
Thank you for giving me the opportunity to revise this paper, for the interest in this work and the helpful suggestions
Reviewer 3
Excellent review. Well done! I do miss one point which could be briefly addressed if the author would like to do so: the possible role of water aerosols in the aspiration of conidia such as the one reported by Warris et al. Molecular epidemiology of Aspergillus fumigatus isolates recovered from water, air, and patients shows two clusters of genetically distinct strains. J. Clin. Microbiol. 2003, 41, 4101–4106 and addressed by Babič, M.N. et al. Fungal Contaminants in Drinking Water Regulation? A Tale of Ecology, Exposure, Purification and Clinical Relevance. Int. J. Environ. Res. Public Health 2017, 14, 636. https://doi.org/10.3390/ijerph14060636
The references have been added with brief sentences about this point.